# A Deep Learning Algorithm for Detecting Acute Pericarditis by Electrocardiogram

**DOI:** 10.3390/jpm12071150

**Published:** 2022-07-15

**Authors:** Yu-Lan Liu, Chin-Sheng Lin, Cheng-Chung Cheng, Chin Lin

**Affiliations:** 1Division of Cardiology, Department of Internal Medicine, Tri-Service General Hospital, National Defense Medical Center, Taipei 114, Taiwan; cylgist80131@gmail.com (Y.-L.L.); littlelincs@gmail.com (C.-S.L.); allexlll@gmail.com (C.-C.C.); 2Graduate Institute of Life Sciences, National Defense Medical Center, Taipei 114, Taiwan; 3School of Medicine, National Defense Medical Center, Taipei 114, Taiwan; 4School of Public Health, National Defense Medical Center, Taipei 114, Taiwan

**Keywords:** artificial intelligence, electrocardiogram, deep learning model, acute pericarditis, ST-segment elevation myocardial infarction

## Abstract

(1) Background: Acute pericarditis is often confused with ST-segment elevation myocardial infarction (STEMI) among patients presenting with acute chest pain in the emergency department (ED). Since a deep learning model (DLM) has been validated to accurately identify STEMI cases via 12-lead electrocardiogram (ECG), this study aimed to develop another DLM for the detection of acute pericarditis in the ED. (2) Methods: This study included 128 ECGs from patients with acute pericarditis and 66,633 ECGs from patients visiting the ED between 1 January 2010 and 31 December 2020. The ECGs were randomly allocated based on patients to the training, tuning, and validation sets, at a 3:1:1 ratio. We used raw ECG signals to train a pericarditis-DLM and used traditional ECG features to train a machine learning model. A human–machine competition was conducted using a subset of the validation set, and the performance of the Philips automatic algorithm was also compared. STEMI cases in the validation set were extracted to analyze the DLM ability of differential diagnosis between acute pericarditis and STEMI using ECG. We also followed the hospitalization events in non-pericarditis cases to explore the meaning of false-positive predictions. (3) Results: The pericarditis-DLM exceeded the performance of all participating human experts and algorithms based on traditional ECG features in the human–machine competition. In the validation set, the pericarditis-DLM could detect acute pericarditis with an area under the receiver operating characteristic curve (AUC) of 0.954, a sensitivity of 78.9%, and a specificity of 97.7%. However, our pericarditis-DLM also misinterpreted 10.2% of STEMI ECGs as pericarditis cases. Therefore, we generated an integrating strategy combining pericarditis-DLM and a previously developed STEMI-DLM, which provided a sensitivity of 73.7% and specificity of 99.4%, to identify acute pericarditis in patients with chest pains. Compared to the true-negative cases, patients with false-positive results using this strategy were associated with higher risk of hospitalization within 3 days due to cardiac disorders (hazard ratio (HR): 8.09; 95% confidence interval (CI): 3.99 to 16.39). (4) Conclusions: The AI-enhanced algorithm may be a powerful tool to assist clinicians in the early detection of acute pericarditis and differentiate it from STEMI using 12-lead ECGs.

## 1. Introduction

Acute pericarditis, a pericardial inflammatory disorder, accounts for approximately 5% of patients with nonischemic chest pain that are admitted to the emergency department (ED) [1]. The etiologies of acute pericarditis could be infectious or noninfectious causes, and the mainstay treatments include nonsteroidal anti-inflammatory drugs (NSAIDs) or aspirin, and specific therapy for the underlying causes if identified [2,3]. The prognosis is generally favorable in the most common viral or idiopathic pericarditis. However, severe complications may accompany this, such as cardiac tamponade and pericardial constriction, which are associated with poor prognosis [1,4,5,6]. Early diagnosis and proper management help to prevent catastrophic outcomes.

The diagnosis of acute pericarditis is established if at least two of the following criteria are met: chest pain compatible with pericarditis, pericardial friction rubs, new widespread ST elevation or PR depression on electrocardiogram (ECG), and new or worsening pericardial effusion [5]. Chest pain is the most common symptom of acute pericarditis, with variable clinical manifestations and severity among patients, which often resembles acute myocardial ischemia and provokes a diagnostic dilemma in clinical practice. Approximately 19–25% of patients with acute pericarditis are mistaken for ST-segment elevation myocardial infarction (STEMI). This incorrect diagnosis may lead to inappropriate thrombolytic therapy and false activation of the cardiac catheterization laboratory, which has several adverse consequences, such as patient distrust, decreased productivity of the medical staff, and unnecessary procedural risk [7,8,9]. Therefore, a diagnostic supportive tool is necessary to improve the accurate diagnosis of acute pericarditis.

A 12-lead ECG remains the most important tool in the diagnosis of acute pericarditis [10]. The classical ECG appearance is widespread ST-segment elevation [5]. From approximately 0.5% to 3.4% of the presumed STEMI patients in the emergency department are later diagnosed with acute pericarditis [11]. Some ECG findings have been proposed to distinguish between STEMI and acute pericarditis. The ST-segment depression in leads other than V1 and aVR, greater ST-segment elevation in lead III than in lead II, the RT checkmark sign, and convex ST-segment elevation are more common in STEMI ECGs. Spodick’s sign and PR-segment depression are more prevalent in acute pericarditis. However, these ECG features lack specificity in clinical applications. Computerized analysis in the 12-lead ECG machines has been widely used and has enhanced the correct ECG interpretation. The automated diagnosis appeared to influence the decision-making of the physicians, but overdiagnosis of STEMI by the automated ECG analysis, which was detected by a simple algorithm, was reported, with a false-positive rate up to 42% with acute pericarditis as one of the leading causes [12,13,14]. As artificial intelligence (AI) techniques rapidly evolved, several deep learning models (DLMs) were developed and shown to achieve the performance of human experts in detecting numerous cardiac diseases [15,16,17,18,19,20,21]. Several AI-based algorithms have been used to detect STEMI; however, to the best of our knowledge, there is no study regarding the application of AI to recognize acute pericarditis [22]. For young faculties, ECG interpretation between these two diseases might be confusing, especially in the rushed environment of an emergency room. We anticipated that the application of DLM in ECG interpretation may help to differentiate acute pericarditis from STEMI and serve as an excellent diagnostic supportive tool for front-line physicians.

We have developed a DLM for STEMI detection (STEMI-DLM) [23]. In this study, we aimed to train and validate a DLM for detecting acute pericarditis by ECG and compared this to the commercial algorithm in standard 12-lead ECG machines. Moreover, we incorporated the previously developed STEMI-DLM and generated a new strategy to improve the diagnostic accuracy and assist physicians in solving this clinical dilemma.

## 2. Method

### 2.1. Study Population

This retrospective cohort study was performed at Tri-Service General Hospital, a single specialist tertiary referral center in northern Taiwan. Ethical approval was obtained through the center’s institutional review board (IRB) (IRB No. C202005055). We included all adult patients (18 years or older) who visited our emergency department with at least 1 digital, standard, 10-s, 12-lead ECG acquired in the supine position between 1 January 2010 and 31 December 2020. The pericarditis cases were primarily identified by ICD-9-CM diagnoses codes of 420. X or ICD-10-CM codes of I30. X, and the corresponding medical records were reviewed by two independent cardiologists. The diagnosis of acute pericarditis was established if at least two of the following criteria were met: chest pain consistent with pericarditis, pericardial friction rubs upon auscultation, typical ECG changes, or pericardial effusion (new or worsening). Patients with correlating ICD codes who did not meet the above criteria were excluded, and the remaining cases without correlating ICD codes were defined as the control group.

For patients with multiple ECGs, each ECG record was defined as the basic unit in this study. A total of 66,761 ECGs from patients aged from 25 to 84 years were included in this study and randomly assigned into three groups: development, tuning, and validation sets. The development set was used to train the DLM and comprised 99 pericarditis ECGs and 39,820 non-pericarditis ECGs. The tuning set was used to optimize the DLM and select the hyperparameters, and contained 10 pericarditis ECGs and 13,767 non-pericarditis ECGs. The validation set was used to test the DLM performance, and consisted of 19 pericarditis ECGs and 13,046 non-pericarditis ECGs.

Since the differential diagnosis between pericarditis and ST-elevation myocardial infarction (STEMI) is challenging in clinical practice, we also identified patients with STEMI by coronary artery acute occlusion in coronary angiography. There were 671, 42, and 49 STEMI ECGs in the development set, tuning set, and validation set, respectively.

### 2.2. Data Source

ECGs in both the pericarditis and the non-pericarditis groups were collected by Philips 12-Lead ECG machines (PH080A, Philips Medical Systems, Andover, MA, USA) with 10 s and a sampling rate of 500 Hz. Since we also trained a machine learning model to compare with the DLM, the 8 quantitative ECG measures and 31 most popular diagnostic pattern classes were also collected. The eight ECG measurements included heart rate, PR interval, QRS duration, QT interval, correct QT interval, P wave axis, RS wave axis, and T wave axis. Data for these variables were 90–100% complete, and missing values were imputed using multiple imputations. The 31 clinical diagnosis patterns were parsed from the structured findings statements based on key phrases that are standard within the Philips system, which included abnormal T wave, atrial fibrillation, atrial flutter, atrial premature complex, complete AV block, complete left bundle branch block, complete right bundle branch block, first degree AV block, incomplete left bundle branch block, incomplete right bundle branch block, ischemia/infarction, junctional rhythm, left anterior fascicular block, left atrial enlargement, left axis deviation, left posterior fascicular block, left ventricular hypertrophy, low QRS voltage, pacemaker rhythm, prolonged QT interval, right atrial enlargement, right ventricular hypertrophy, second degree AV block, sinus bradycardia, sinus pause, sinus rhythm, sinus tachycardia, supraventricular tachycardia, ventricular premature complex, ventricular tachycardia, and Wolff–Parkinson–White syndrome [24]. The above features were used to develop the machine learning model described in the next section.

We collected patient characteristics from electronic medical records with follow-up information. The corresponding laboratory data within 3 days before enrollment were assigned to each ECG. Baseline comorbidities were extracted by ICD-9 and ICD-10 codes [23,24,25,26]. Admission within 3 days was the outcome of interest, where CV-caused hospitalization was defined by an individual cardiologist, making any cardiac disease the major diagnosis.

### 2.3. The Implementation of the Deep Learning Model

We previously developed an 82-layer convolutional neural network called ECG12Net, and the technological details were provided in previous studies [25]. Based on the same architecture, a new DLM was developed for the detection of acute pericarditis, which required about 15–20 s from inputting the ECG signals to generating the interpretation results. To compare the use of ECG voltage–time traces and the corresponding, clinically reported ECG measures, we trained an Xtreme gradient-boosting model (XGB model) using 31 diagnostic pattern classes and 8 ECG measurements to recognize pericarditis in the training set. The XGB model used gradient-boosted decision trees to calculate the loss function and provides excellent computational speed and accuracy in terms of both time and prediction [27]. It displayed the following advantages: (1) effectively handling missing values; (2) preventing overfitting; (3) reducing computation time using parallel and distributed computation.

### 2.4. Human–Machine Competition

We held a human–machine competition to evaluate the performance of our DLM. The database used 87 ECGs sampled from the validation set, including 17 pericarditis cases and 70 non-pericarditis cases. Seven doctors participated in the competition (two internal medicine residents, two emergency medicine residents, one emergency physician, and two cardiologists), and all the doctors completed the tests using an online standardized data entry program without any patient information except the ECGs. In addition, the Philips 12-lead algorithm was also included to detect pericarditis in the competition [28]. We calculated the sensitivity and specificity of the doctors for comparison with those of the DLM.

### 2.5. Statistical Analysis

The characteristics and laboratory results were presented as the means and standard deviations for continuous variables and as numbers and percentages for categorical variables, respectively. We used Student’s *t*-test or the chi-square test to compare the results between the two groups, as appropriate, and *p* values < 0.05 were considered statistically significant. The statistical analysis was performed with R version 3.4.4, and the package MXNet version 1.3.0 was used to implement our DLM [29].

In the primary analysis, we compared the performance of our DLM to human experts, Philips 12-lead algorithm, and the XGB model. Receiver-operating characteristic (ROC) curves and areas under the curve (AUCs) were applied to evaluate the performance of pericarditis recognition of DLMs and machine-learning algorithms. The operating point was selected based on the maximum Youden’s index derived from the tuning set. To identify the relationship between clinical characteristics and pericarditis, and which of these characteristics leads to misdiagnosis by DLMs; logistic regression was applied to calculate the odds ratios (ORs) of each clinical characteristic.

In the secondary analysis, we also included the DLM to recognize STEMI that was previously developed in the pericarditis identification process [30]. We analyzed the non-pericarditis ECGs in the validation set: “false-positive” cases that were identified and “true negative” cases that were not identified were stratified by this process. Kaplan–Meier survival analysis was performed with the available follow-up data, which were stratified by the DLM predictions on each outcome of interest. The data were censored based on the most recent encounter. The hazard ratios (HRs) were calculated through the Cox proportional hazard model, and the values with 95% confidence intervals (95% CIs) were reported for all data.

## 3. Results

The patient characteristics in the development, tuning and validation sets are shown in Table 1. Patients with pericarditis in all the datasets were younger, had a higher estimated glomerular filtration rate, had a higher hemoglobin level, and had fewer comorbidities, such as diabetes mellitus, hypertension, and hyperlipidemia. All pericarditis cases and from 16.8% to 19.2% non-pericarditis cases had chest pain.

Figure 1 shows the performance of our DLM in three subsets. In the human–machine competition, the AUC of the DLM used to detect pericarditis was 0.943, with a corresponding sensitivity of 76.5% and a specificity of 100.0%. The DLM had the best performance compared to all human experts, Philips automatic ECG interpretation, and the XGB model. The specificities of Philips automatic ECG interpretation and human experts were high, but the sensitivities were much lower than our DLM in terms of pericarditis detection. The AUCs were 0.954 and 0.952 in the validation dataset and chest pain subset, respectively, with the same sensitivities of 78.9% and similar specificities of 97.7% and 97.6%, which were significantly better than the XGB model and Philips automatic ECG interpretation. The consistency analysis in the human–machine competition is demonstrated in Appendix A. We shared two representative ECGs of acute pericarditis. The ECG in Appendix A revealed sinus rhythm with an inverted T wave, which was precisely recognized by our DLM as acute pericarditis ECG but misidentified as non-pericarditis ECG by all participating physicians. The ECG in Appendix A showed sinus tachycardia with diffuse ST-segment elevation and PR-segment elevation in lead aVR, which was correctly identified by all physicians but not the DLM.

The predictive abilities of the individual lead are shown in Appendix A, and there was no single lead showing a better performance than the integration of all 12 leads. We tried to enhance the predictive accuracy using additional patient characteristics, and Appendix A shows that male sex, younger age, a history of coronary artery disease, and non-diabetes contributed to a significant risk of pericarditis. We then incorporated these risk factors into our DLM. However, the diagnostic value of the integration model was not better than a simple DLM using ECG alone, as shown in Appendix A.

Since false-positive prediction by DLM may identify a “previvor” of cardiovascular diseases [31], Appendix A analyzed the association between patient characteristics and DLM predictions in non-pericarditis patients. The most dominant characteristic was STEMI, which is easily confused with acute pericarditis in clinical practice and has the highest association (OR: 4.97, 95% CI: 1.96–12.62). Therefore, we evaluated the prediction abilities for both diseases between our pericarditis DLM and the previously developed STEMI-DLM [23]. In 19 patients with pericarditis, 14 of the ECGs were simultaneously identified as STEMI and pericarditis, and 3 of them were misidentified by both DLMs. Only two inconsistent predictions implied that STEMI-DLM can initially identify potential pericarditis cases. In 49 patients with STEMI, STEMI-DLM identified 44 with a sensitivity of 89.8%, and pericarditis-DLM correctly identified 39 of them as non-pericarditis. We considered merging two DLMs as a new strategy for differential diagnosis between STEMI and pericarditis. All ED cases were initially analyzed by STEMI-DLM, and the ECGs with a high STEMI likelihood were reanalyzed by pericarditis-DLM for further diagnosis. Figure 2C shows a sensitivity of 73.7% and a specificity of 99.4% using this new strategy for identifying pericarditis, which significantly improved the positive predictive value in the validation set to 14.4%. Figure 2D shows the results in the chest pain subset. Due to the similar sensitivity and specificity but higher prevalence of pericarditis, the positive predictive value was further increased to 50.0%, which emphasized the clinical impact of this new strategy.

We analyzed the clinical outcomes in non-pericarditis patients who were misidentified using the strategy mentioned in Figure 3. The risk of CV-caused hospitalization within 3 days in the DLM-identified group was significantly higher than that in the DLM-unidentified group, with an HR of 3.13 (95% CI: 1.67–5.85). Importantly, this risk difference is evident on the first day. In contrast, the risk of non-CV-caused hospitalization was similar between the DLM-identified and DLM-unidentified groups. This risk difference was more pronounced in patients with chest pain, with an HR of 8.09 (95% CI: 3.99–16.39).

## 4. Discussion

To the best of our knowledge, machine learning and deep learning are not used to detect acute pericarditis, and we described the first DLM for the detection of acute pericarditis. The AUC of the DLM in detecting acute pericarditis was 0.94, and the DLM performance was superior to that of cardiologists, emergency physicians, and Philips automatic ECG interpretation. Of note, we proposed a new AI-based strategy, integrating STEMI-AI and our pericarditis-AI to evaluate patients with acute chest pain at the ED. Intriguingly, among non-pericarditis patients, those who were misidentified as pericarditis by this strategy had a higher risk of hospitalization related to other cardiac disorders compared to those who were correctly diagnosed. Our DLM for acute pericarditis detection provides both decision support and prognosis prediction.

In the human–machine competition, the DLM had a better performance than the human experts and the Philips algorithm, with both high sensitivities and specificities for detecting acute pericarditis. Possible reasons for the low sensitivities of human experts in detecting acute pericarditis include diverse ECG presentations and disease stages of acute pericarditis. The typical ECG pattern of acute pericarditis is a generalized, concave-upward ST-segment elevation with PR-segment depression, which was only exhibited in less than 60% of the cases [5,32]. In addition, the ECGs in acute pericarditis differ over time, with a four-stage evolution. The classic manifestation of diffuse ST-segment elevation and PR-segment depression is present in stage 1. The J points return to baseline, and T waves start to flatten in stage 2. T waves are inverted in almost all leads in stage 3, and the generalized T-wave inversion is gradually resolved in stage 4, with focal T-wave flattening or inversion persisting in rare cases [1,33,34]. Interestingly, in most circumstances, only the ECG appearance in stage 1 can be recognized by the physicians and the Philips automatic ECG interpretation. The difference in diagnostic sensitivity implied the probably undiscovered ECG characteristics of acute pericarditis that were identified by our DLM.

In real-world practice, acute pericarditis and STEMI have similar presentations, including acute chest pain and ST-segment elevation on ECG, which lead to challenges in differential diagnosis. Figure 4 emphasized the different mechanisms of acute pericarditis and STEMI, which resulted in the distinct management of the two disorders. The management of acute pericarditis and STEMI is distinct. Supportive treatment with medications of anti-inflammatory therapy with either NSAID or aspirin and colchicine is recommended to shorten the symptom persistence and reduce the recurrence rate [1,5,10]. On the other hand, immediate coronary angiography with a timely reperfusion strategy is essential to preserve the myocardium in STEMI patients [35,36,37]. Many of the AI-enabled ECG studies were devoted to STEMI identification to promptly activate the PCI team and reduce the door-to-balloon time. To date, this is the first DLM study to distinguish acute pericarditis from STEMI. In our study, STEMI-DLM failed to differentiate STEMI from acute pericarditis by ECGs in patients with acute pericarditis, with a rate of misdiagnosis up to 78.9%. Importantly, our pericarditis-DLM successfully recognized STEMI ECGs as non-pericarditis cases. Accordingly, we developed a new AI-based strategy by integrating our STEMI-DLM and pericarditis-DLM using a two-step process to evaluate patients with acute chest pain. The first step consists of screening the raw ECGs using our STEMI-DLM. This step allows for us to include the ECGs of both STEMI and acute pericarditis with similar sensitivities in both DLMs to detect acute pericarditis, and to use the much higher sensitivity of STEMI-DLM to identify STEMI ECGs. Then, in the second step, the included ECGs were analyzed by the pericarditis-DLM and were divided into acute pericarditis ECGs and non-pericarditis ECGs. The latter are the true STEMI ECGs. This new strategy is capable of both detecting acute pericarditis and discriminating it from STEMI ECGs in chest pain patients with high sensitivity and positive predictive values.

We noticed a substantial portion of false-positive results in non-pericarditis patients. Interestingly, compared to the non-pericarditis cases, who were accurately diagnosed by our strategy, these patients with false-positive results had an eight-fold increased risk of hospitalization due to other cardiac disorders that occurred within 3 days of the index ECGs. Although the underlying mechanism is not well-understood, AI-ECG models have demonstrated the possibility of predicting the risk of certain diseases and clinical outcomes in the future by learning hidden ECG signals [31,38,39,40,41]. False-positive cases carry a higher risk of arrhythmia, leading to poorer prognosis compared with true-negative cases [42]. Our findings supported the existence of unknown ECG features preceding the onset of clinical symptoms. Accordingly, in a patient who was initially identified as having acute pericarditis using our strategy, we suggest a careful cardiovascular evaluation even if acute pericarditis is later excluded.

Our pericarditis-DLM exhibits several powerful applications. Current ECG features differentiating STEMI from acute pericarditis have low specificity in clinical scenarios [43]. Our pericarditis-DLM correctly diagnosed 78.9% of pericarditis cases and identified 89.8% of STEMI cases as non-pericarditis ECGs, which may effectively help the differential diagnosis between these two diseases in the ED. Moreover, our AI-based strategy may be useful in rural health services, where experienced physicians are lacking, particularly in situations where decisions must be made regarding transfers to a PCI available center. Additionally, acute pericarditis emerged as an important disorder during the COVID-19 pandemic. The incidence was reported to be approximately 1.5% in COVID-19 patients [44]. Unlike other viral etiologies, pericarditis related to COVID-19 is associated with a poor prognosis compared to pericarditis without cardiac involvement in either hospitalized or non-hospitalized patients. Acute pericarditis after COVID-19 vaccination is another prominent issue, with an overall incidence of from 1.88 to 13.5 cases per million doses, which affects older patients after either the first or second immunization [45,46]. The application of telemedicine technologies and virtual software is shown to provide a promising opportunity to reduce ED visits, preserves health care resources, and prevents direct transmission of the pathogen [47]. Equipping wearable devices with algorithms can help to detect pericarditis cases in the COVID-19 pandemic era (Central Illustration). All the evidence highlights the unmet clinical needs of our pericarditis-DLM.


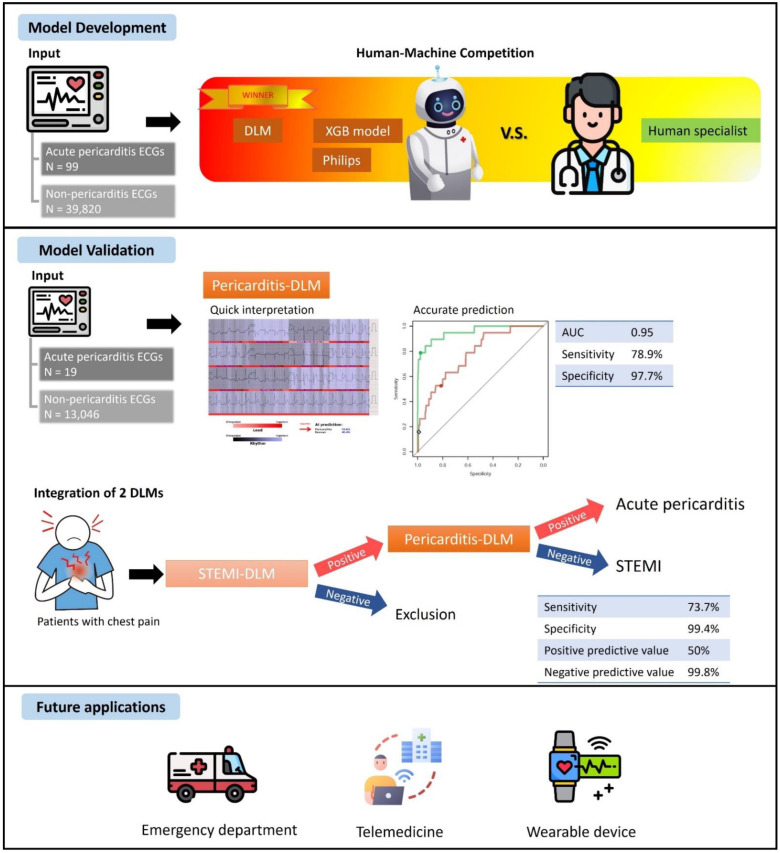
**Central Illustration.** The flow diagram of the development, validation, and future application of the pericarditis deep learning model (DLM). The pericarditis-DLM was developed and trained by 39,919 ECGs. The DLM had the best performance compared to the built-in algorithm of ECG machine, another machine-learning model, and human specialists. The DLM achieved a good diagnostic ability, with an AUC of 0.95, a sensitivity of 78.9% and a specificity of 97.7%. The pericarditis-DLM and the previously developed STEMI-DLM were combined to a new strategy, which exhibited an excellent power to differentiate between acute pericarditis and STEMI, with a sensitivity of 73.7% and a positive predictive value of 50%. The applications of our DLM in emergency department, telemedicine and wearable devices may possibly help to improve the healthcare and outcomes of cardiovascular diseases in the future.

### Limitation

Several limitations of this study should be acknowledged. First, this is a retrospective single-center study. Since the incidence of acute pericarditis was low, pericarditis-DLM may perform better using a larger number of patients. Due to the limited number of patients in this research, we did not further classify the ECGs according to the stages of acute pericarditis and calculate the AUCs. Second, the AI models in our study require further external validation. Future studies are required to validate the performance of our pericarditis-DLM in other populations. Third, the algorithms in our study were primarily developed by analyzing ECG signals, whereas clinical symptoms and signs are the fundamentals for diagnosing acute pericarditis and differentiating it from STEMI. A focused history and a careful physical examination are required to overcome this limitation of our pericarditis-DLM. Fourth, patients with chronic pericarditis were not included in this study. Fifth, information regarding the history of recent infection, fever symptoms, and CRP levels was not available in our research, despite the potential of increasing differentiation between acute pericarditis and STEMI. Hence, our pericarditis-DLM provides diagnostic support, which should not be used as the sole evidence when making a diagnosis. Finally, the methodological drawbacks of the existing DLMs are unknown during the process of DLM interpretation [48]. Even with these limitations, our integrated pericarditis-DLM and STEMI-DLM systems provide novel information for patients with acute chest pain at the ED.

## 5. Conclusions

This is the first DLM using ECG signals to detect acute pericarditis. We further developed a new AI-based strategy by incorporating the pericarditis-DLM and the STEMI-DLM to discriminate acute pericarditis from STEMI ECGs in patients with acute chest pain. Furthermore, patients with false-positive results using the integrated strategy carry a higher risk of hospitalization due to cardiac disorders, and should be comprehensively evaluated. Although large-scale, population-based studies are needed, our pericarditis-DLM is definitely a promising diagnosis support tool to detect pericarditis in the ED, and could be applied for telemedicine and wearable technologies in the COVID-19 pandemic era.

## Figures and Tables

**Figure 1 jpm-12-01150-f001:**
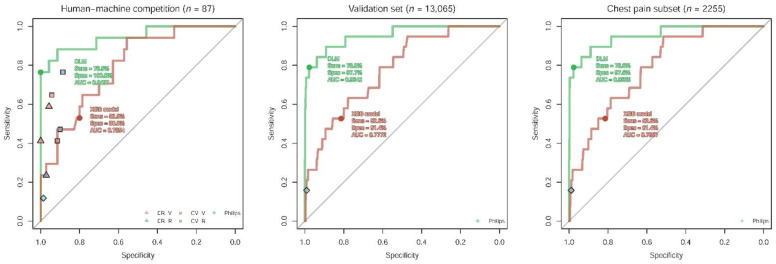
Summary of model performance as the area under the receiver operating characteristic curve for predicting pericarditis. The ROC curves were made by the predictions of the deep learning model (DLM) using raw ECG signals and the XGB model integrating ECG measures (8 numerical values and 31 diagnostic labels), respectively. Each point represents the performance of humans and Philips automatic ECG interpretation. The cut points of the DLM and XGB model were based on Youden’s index in the tuning set.

**Figure 2 jpm-12-01150-f002:**
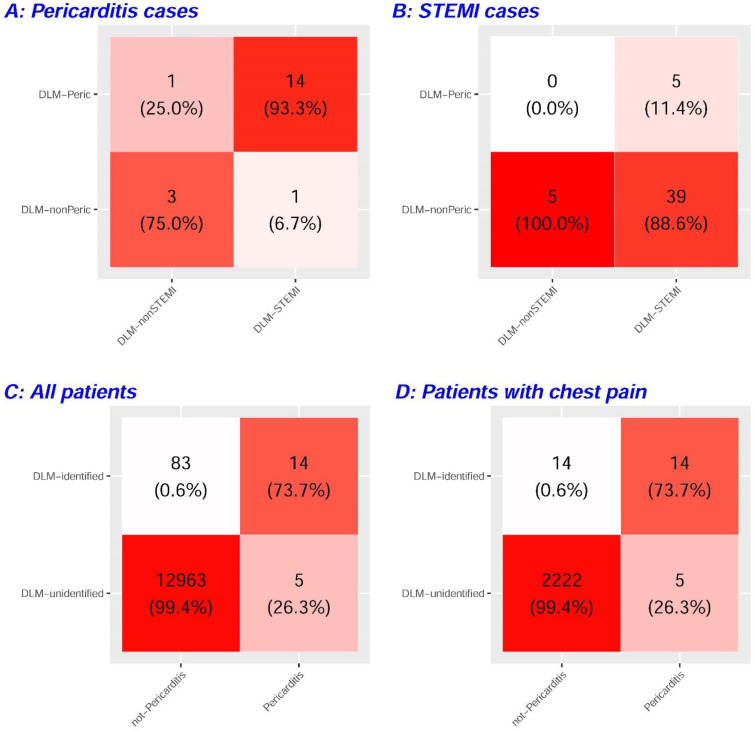
Integration of our pericarditis deep learning model (DLM) and previous STEMI DLM in the validation set. (**A**) The cross-table of predictions of two DLMs in pericarditis cases. (**B**) The cross-table of predictions of two DLMs in STEMI cases. (**C**) DLM identification was defined as the intersection of DLM-pericarditis and DLM-STEMI, which was defined as a new strategy to identify potential pericarditis cases. (**D**) The same strategy was applied to patients with chest pain.

**Figure 3 jpm-12-01150-f003:**
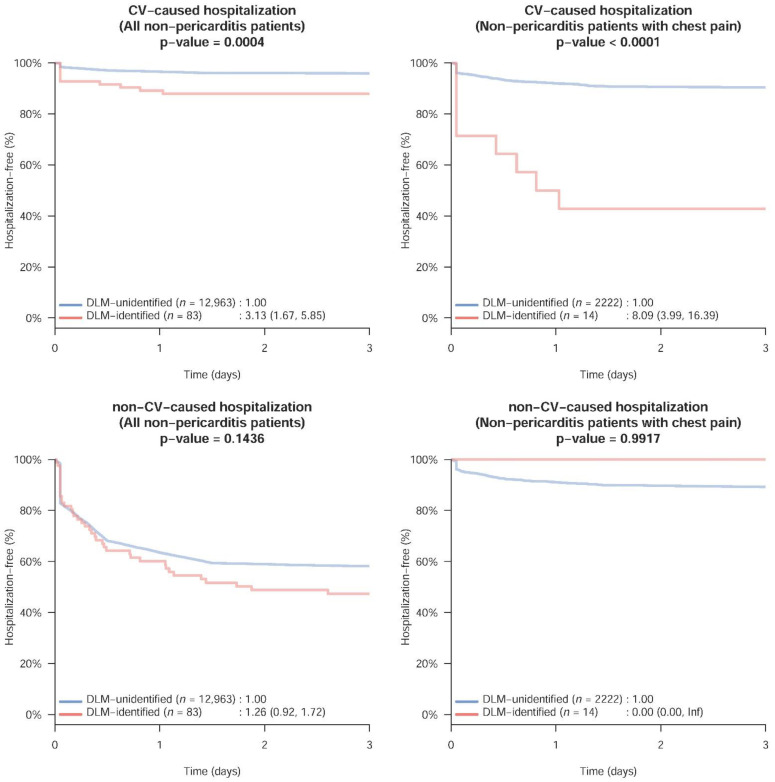
3-day CV- and non-CV-caused hospitalization in non-pericarditis cases stratified by DLM classification. DLM identification was defined as the intersection of DLM-pericarditis and DLM-STEMI. A higher risk of 3-day CV-caused hospitalization was present when the DLM defined the ECG as abnormal compared with those who were classified as having a normal ECG by DLM. The numbers reported in the legend are the hazard ratios.

**Figure 4 jpm-12-01150-f004:**
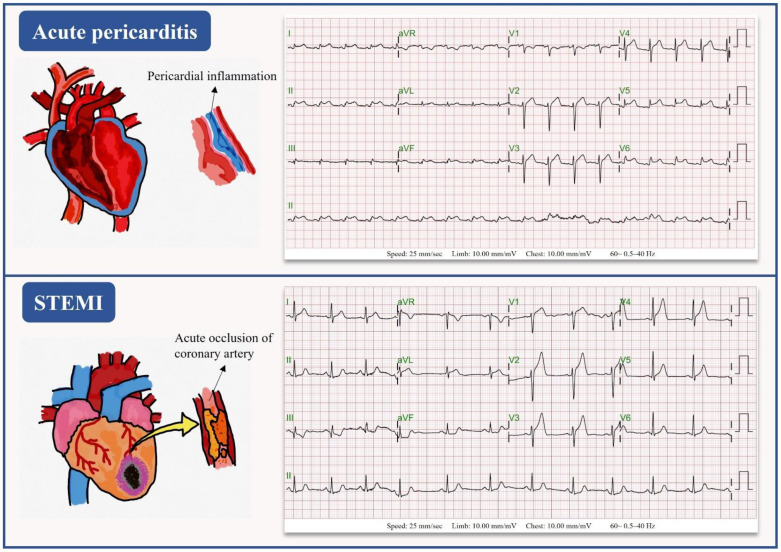
Different impacts of acute pericarditis and STEMI in the physical heart. Inflammation of pericardium resulted in acute pericarditis (**upper** panel), whereas STEMI is caused by the acute total occlusion of epicardial coronary arteries (**bottom** panel).

**Table 1 jpm-12-01150-t001:** Corresponding patient characteristics of pericarditis and non-pericarditis visits in each dataset.

	Development Set			Tuning Set			Validation Set		
	Pericarditis (*n* = 99)	Non-Pericarditis (*n* = 39,820)	*p* Value	Pericarditis (*n* = 10)	Non-Pericarditis (*n* = 13,767)	*p* Value	Pericarditis (*n* = 19)	Non-Pericarditis (*n* = 13,046)	*p* Value
Clinical features									
Chest pain	99 (100.0%)	7637 (19.2%)	<0.001	10 (100.0%)	2316 (16.8%)	<0.001	19 (100.0%)	2236 (17.1%)	<0.001
STEMI	6 (6.1%)	665 (1.7%)	0.007	0 (0.0%)	42 (0.3%)	1.000	0 (0.0%)	49 (0.4%)	1.000
Demographic data									
Gender (male)	81 (81.8%)	20,983 (52.7%)	<0.001	7 (70.0%)	7002 (50.9%)	0.344	15 (78.9%)	7058 (54.1%)	0.030
Age (years)	43.9 ± 18.7	62.1 ± 19.6	<0.001	35.9 ± 1.1	66.0 ± 18.8	<0.001	51.7 ± 22.9	66.2 ± 18.4	0.007
BMI (kg/m^2^)	24.4 ± 4.3	24.3 ± 5.8	0.977	22.9 ± 3.1	24.3 ± 6.5	0.491	26.5 ± 3.2	24.4 ± 6.0	0.056
Disease history									
AMI	11 (11.1%)	2136 (5.4%)	0.011	0 (0.0%)	725 (5.3%)	1.000	0 (0.0%)	833 (6.4%)	0.630
Stroke	7 (7.1%)	6697 (16.8%)	0.010	0 (0.0%)	3205 (23.3%)	0.130	1 (5.3%)	3622 (27.8%)	0.029
CAD	28 (28.3%)	9828 (24.7%)	0.407	3 (30.0%)	4241 (30.8%)	1.000	6 (31.6%)	4871 (37.3%)	0.604
HF	0 (0.0%)	3568 (9.0%)	0.002	0 (0.0%)	2076 (15.1%)	0.377	1 (5.3%)	2489 (19.1%)	0.152
AF	0 (0.0%)	2722 (6.8%)	0.007	0 (0.0%)	1401 (10.2%)	0.613	1 (5.3%)	1299 (10.0%)	1.000
DM	6 (6.1%)	9387 (23.6%)	<0.001	0 (0.0%)	4442 (32.3%)	0.037	1 (5.3%)	4900 (37.6%)	0.004
HTN	19 (19.2%)	15,111 (37.9%)	<0.001	0 (0.0%)	7008 (50.9%)	0.001	6 (31.6%)	7284 (55.8%)	0.033
CKD	1 (1.0%)	4512 (11.3%)	0.001	0 (0.0%)	2795 (20.3%)	0.229	1 (5.3%)	3047 (23.4%)	0.098
HLP	16 (16.2%)	11,463 (28.8%)	0.006	4 (40.0%)	4984 (36.2%)	0.755	0 (0.0%)	5144 (39.4%)	<0.001
COPD	15 (15.2%)	6533 (16.4%)	0.736	3 (30.0%)	3314 (24.1%)	0.712	1 (5.3%)	3581 (27.4%)	0.030
Laboratory test									
eGFR (ml/min)	97.4 ± 32.1	77.2 ± 39.5	<0.001	94.9 ± 9.0	70.4 ± 40.6	0.010	123.6 ± 74.7	69.6 ± 42.5	<0.001
Cr (mg/dL)	1.0 ± 0.8	1.5 ± 1.9	0.016	0.9 ± 0.2	1.8 ± 2.3	0.251	0.8 ± 0.2	1.9 ± 2.4	0.006
BUN (mg/dL)	16.3 ± 8.8	24.9 ± 22.3	0.003	11.3 ± 3.5	28.1 ± 24.6	<0.001	13.9 ± 4.9	28.6 ± 25.8	0.001
Na^+^ (mmol/L)	135.8 ± 3.3	136.7 ± 5.1	0.101	135.2 ± 1.5	136.4 ± 5.1	0.049	135.8 ± 2.5	136.3 ± 5.5	0.144
K^+^ (mmol/L)	3.8 ± 0.5	3.9 ± 0.6	0.166	4.2 ± 0.7	4.0 ± 0.7	0.461	4.0 ± 0.5	4.0 ± 0.7	0.696
Cl^−^ (mmol/L)	101.4 ± 4.4	102.6 ± 5.9	0.202	100.8 ± 2.3	102.2 ± 5.8	0.128	107.6 ± 3.1	102.0 ± 6.2	0.012
tCa^++^ (mg/dL)	8.6 ± 0.6	8.5 ± 0.7	0.769	8.4 ± 0.3	8.6 ± 0.8	0.724	8.2 ± 0.4	8.6 ± 0.7	0.014
Mg^++^ (mg/dL)	1.9 ± 0.3	2.1 ± 0.4	0.029	2.2 ± 0.2	2.1 ± 0.4	0.103	2.0 ± 0.0	2.1 ± 0.4	0.528
TnI (pg/mL)	1912.7 ± 3393.7	607.7 ± 5459.5	0.049	125.6 ± 109.7	240.4 ± 2545.5	0.027	1073.6 ± 3789.5	245.5 ± 2863.0	0.622
CK (U/L)	217.7 ± 226.0	222.1 ± 811.5	0.965	69.6 ± 15.4	186.7 ± 766.2	0.160	181.2 ± 210.7	168.1 ± 712.5	0.104
BNP (pg/mL)	361.1 ± 415.1	528.4 ± 938.9	0.314	33.2 ± 28.5	569.7 ± 987.0	0.015	144.3 ± 39.3	673.1 ± 1120.2	0.642
GLU (gm/dL)	115.2 ± 18.5	150.3 ± 88.0	0.024	109.2 ± 24.0	149.2 ± 80.9	0.109	120.0 ± 20.7	151.8 ± 89.4	0.258
Hb (g/dL)	13.9 ± 2.1	12.7 ± 2.4	<0.001	14.8 ± 2.2	12.3 ± 2.5	0.002	14.1 ± 1.9	12.1 ± 2.5	0.001
WBC (10^3^/uL)	11.8 ± 4.3	9.5 ± 6.2	0.002	10.3 ± 6.0	9.3 ± 4.7	0.660	12.9 ± 5.5	9.2 ± 7.0	0.001
PLT (10^3^/uL)	216.8 ± 72.7	238.0 ± 90.4	0.047	281.2 ± 64.6	233.8 ± 92.0	0.037	200.0 ± 49.2	230.3 ± 95.2	0.144
AST (U/L)	43.1 ± 46.0	52.6 ± 174.4	0.632	29.9 ± 18.7	40.5 ± 117.2	0.652	25.3 ± 20.6	44.3 ± 137.8	0.083
ALT (U/L)	60.6 ± 123.1	36.1 ± 126.7	0.180	40.3 ± 37.7	31.1 ± 81.4	0.804	23.6 ± 8.5	34.3 ± 122.6	0.141
TG (gm/dL)	108.2 ± 35.8	126.0 ± 142.1	0.532	235.0 ± 0.0	122.2 ± 146.6	0.030	69.6 ± 14.6	120.8 ± 132.4	0.041
TC (gm/dL)	135.9 ± 32.3	153.0 ± 48.4	0.003	174.1 ± 22.1	149.6 ± 48.8	0.016	137.0 ± 16.3	148.0 ± 45.6	0.416

Abbreviations: STEMI, ST elevation myocardial infarction; BMI, body mass index; AMI, acute myocardial infarction; CAD, coronary artery disease; HF, heart failure; AF, atrial fibrillation; DM, diabetes mellitus; HTN, hypertension; CKD, chronic kidney disease; HLP, hyperlipidemia; COPD, chronic obstructive pulmonary disease; eGFR, estimated glomerular filtration rate; Cr, creatinine; BUN, blood urea nitrogen; Na^+^, sodium; K^+^, potassium; Cl^−^, chloride; tCa^++^, total calcium; Mg^++^, magnesium; TnI, troponin I; CK, creatine kinase; BNP, brain natriuretic peptide; GLU, fasting glucose; Hb: hemoglobin; WBC, white blood cell count; PLT, platelet; AST, aspartate aminotransferase; ALT, alanine aminotransferase; TG, triglyceride; TC, total cholesterol.

## Data Availability

The data presented in this research are available upon formal request from the corresponding author.

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
