# Peer review of "A Deep Learning Algorithm for Detecting Acute Pericarditis by Electrocardiogram"

_jpm, 2022, doi:10.3390/jpm12071150_

Round 1

Reviewer 1 Report

Dear Authors and Editorial Office, 

Thank you for giving me the opportunity  of reviewing the paper on the subject of AI/DLM in acute pericarditis and anterior STEMI. The authors highlight correctly, that the differential diagnosis of the diseases mentioned above are not alway easy, and misdiagnosis, especially STEMI, can lead to increased mortality.

Questions/comments:

1. The paper does not state the time frames of the DLM to provide diagnosis. It is very important to have the correct diagnosis in a timely manner, especially in STEMI.

2. With the regards to the stages of pericarditis, did you calculate the AUC-s for each?

3. Would chronic pericarditis have the same AUCs?

4. Some patient characteristics are integrated into the algorithm, as I understood, causing no changes in efficacy. Would the terms "infection in the last two weeks", "age below 30" (very low chance of STEMI), CRP 5 x above normal range" or "fever" would increase the success when differentiating between pericarditis and STEMI?

Reviewer 2 Report

The purpose of the manuscript is to describe the implementation of diagnosing acute pericarditis using a deep learning model and differentiate it from STEMI. The manuscript used analysis using a previously published deep learning technique by the same author. Although the detection of acute pericarditis seems to be a success but differentiating acute pericarditis and STEMI lacks confidence due to a low number of case/measurements/samples. The authors are suggested to work with the major/minor issues before the manuscript is suitable for publication in the “Journal of Personalized Medicine”.

Major points:

1.      To distinguish pericarditis from STEMI: A. ST depression in a lead other than aVR or V1, if yes then STEMI. B. Is there convex up or horizontal ST elevation, if yes then STEMI. C. If ST elevation in lead III is greater than lead II then STEMI. So, based on those information STEMI can be diagnosed by carefully checking ECG in all leads, so why there is the necessity for a diagnostic supportive tool to improve accuracy or differentiate between pericarditis and STEMI.

2.      Discussion related to the use of machine learning and deep learning to detect acute pericarditis need to be included and should be placed before proposing a new technique.

3.      While differentiating pericarditis and ST-elevation-related MI (STEMI) the dataset involves 671, 42, and 49 STEMI ECGs. Do authors think the number of ECGs is sufficient to make a reliable and robust decision about differentiating pericarditis and STEMI?

4.      There must be diagrams pointing out the implication of pericarditis and STEMI in the physical heart.

5.      If possible, authors are requested to provide demographic and physical information of the participants using the range, for example: how many patients or measurements from patients within a certain age range.

6.      What is the reason behind choosing the XGB model to compare? Why not another deep learning model?

7.      The whole experiment needs to be depicted using a flow diagram to give the reader an easy overview.

8.      What is a probable solution for the third limitation listed by the authors?

9.      In figure 2 the cross tables in A and B contains a very small number of cases. Does the author think the numbers are sufficient? Even changing a single number by one would change the percentage drastically.

Minor points:

1.      In the ‘method’ section, ‘data source’ subsection: “included abnormal T wave … … Wolff-Parkinson-White syndrome”- all these 8 lines are directly copied from another article without using any reference. Even though that article was authored by the same author from this manuscript, the citation is a must.

2.      In the ‘method’ section, and ‘statistical analysis’ subsection: the first five lines are also copied without using any reference.

3.      “Approximately 0.5% to 3.4% of the presumed STEMI patients in the emergency department are later diagnosed with acute pericarditis”- why this single statement requires 8 references? 

Round 2

Reviewer 2 Report

The authors have answered all the queries satisfactorily and revised them accordingly.